# Yah's Exemplary Soldiers: African Hebrew Israelites in the Israel Defense Forces

**Andrew Esensten**

Independent Researcher, CA, USA; aesensten@gmail.com

**Abstract:** This article considers the process of identity formation among soldiers in the Israel Defense Forces (IDF) who were born into the African Hebrew Israelites of Jerusalem (AHIJ), more commonly known as the Black Hebrews. The AHIJ are a sect of African Americans who began settling in Israel in 1969 and who identify as direct descendants of the Biblical Israelites. Due to the group's insular nature, the IDF is the primary state institution in which they fully participate, and their mandatory service is a source of both pride and consternation for community members and leaders. Considering the personal experiences of 14 African Hebrew soldiers who enlisted between 2009 and 2010, the article argues that while the soldiers by and large maintain their distinctive identity during the course of their service, they also internalize some of the language, attitudes, and cultural touchstones of the majority Israeli Jewish population. As a result, they experience a kind of "double consciousness", the feeling of dislocation first described by the African American scholar W. E. B. Du Bois at the turn of the twentieth century.

**Keywords:** African Hebrew Israelites; Black Hebrews; Israel; Israel Defense Forces; identity

---

## 1. Introduction

Military service is an important rite of passage for non-Haredi Jewish Israelis, a demographic that is conscripted into the Israel Defense Forces (IDF) for two to three years upon graduating from high school. After completing their service, many veterans continue to serve in the reserves until middle age and are afforded special rights and privileges within society. The IDF is consistently recognized as the most trusted Israeli institution, and the importance of serving in the IDF and defending the State of Israel may very well be the only issue that has full consensus among non-Haredi Jewish Israelis (Reuven 1986).

For the African Hebrew Israelites of Jerusalem (AHIJ), however, military service is a more complicated matter. The AHIJ is the largest intentional community of African American expatriates in the world, with a membership of approximately 3000 spread across Israel but concentrated in the Negev desert towns of Dimona, Arad, and Mitzpe Ramon. While Israelis often speak of them dismissively as a cult, the AHIJ are best described as a spiritual community who were led for many years by a charismatic leader, Ben Ammi Ben Israel (1939–2014). The African Hebrews identify as descendants of the Biblical Israelites and consider Israel to be their ancestral homeland, which they locate in northeastern Africa, not the Middle East (Markowitz 1996). They do not practice Judaism, nor are they recognized as Jews by Israel's religious authorities, but they worship the God of Israel—whom they refer to as Yah, a shortened vocalization of the Tetragrammaton—and adhere to a Torah-based lifestyle.[1] As non-Jews, their social status is inferior to that of Israeli-born Jews and Jewish immigrants from other countries,

---

[1] Much like Karaites, a medieval Jewish sect, the African Hebrews abide by the "written law" of the Torah but reject the "oral law" of Rabbinic Judaism, which includes the Talmud and Midrash.

making their participation in the IDF even more crucial for gaining a measure of social acceptance and economic mobility in Israeli society.

The first African Hebrew youth enlisted in the IDF in July 2004 following a government decision to upgrade community members' status from temporary residency to permanent residency, a step below full citizenship. This upgrade meant, among other things, that all AHIJ youth would need to register for the draft.[2] Since 2004, more than 350 male and female African Hebrew teenagers have been inducted into the IDF and served the country they call home. Community representatives boast an enlistment rate of 100%, though it is probably slightly lower than that. (Over the years, a small number of African Hebrews have deferred their service for health or other reasons.) Many of these soldiers have fought on the front lines in Israel's recent wars, and one has died while on active duty, though not in a combat-related incident.[3]

Despite their enthusiastic participation in this most Israeli of institutions, the AHIJ remain on the periphery of society. Fewer than 200 African Hebrews have Israeli citizenship; the vast majority are American citizens who cannot vote in national elections. On average, fewer than half of the members of each high school class attend university upon graduation, though African Hebrew high school students consistently score high on their exit exams, according to school officials. Those who work outside of their community framework have low-paying jobs in construction, agriculture, textiles, and the service industries. Israeli politics and culture mostly do not interest them. Serving in the military is a major step toward fully joining Israeli society, but is that the community's ultimate goal?

This is the first article to consider military service as a crucial step in the African Hebrews' cautious integration into mainstream Israeli society. Taking cues from previous studies on the military service of ethnic and religious minorities—including Mizrachi Jews (Levy 1998), Russian Jews (Lomsky-Feder and Rapoport 2001; Lomsky-Feder and Ben-Ari 1999, 2007), Ethiopian Jews (Shabtay 1995, 1999), Haredi Jews (Stadler and Ben-Ari 2003), Druze (Frisch 1993; Atashi 2001), Circassians (Geller 2012), and Arabs (Kanaaneh 2003, 2005)—this article explores the effects of military service on identity formation among third-generation African Hebrews.[4] Most of the studies cited above focus on the "macro" point of view, that is, how the structure and policies of the IDF shape the identities of soldiers from specific ethnic groups (Kachtan 2012). This article focuses on the "micro" point of view by exploring soldiers' personal, idiosyncratic experiences.

Based on their testimonies, I argue that African Hebrews who serve in the IDF experience a kind of "double consciousness", which the African American scholar W. E. B. Du Bois described as a "peculiar sensation . . . of always looking at one's self through the eyes of others, of measuring one's soul by the tape of a world that looks on in amused contempt and pity" (Du Bois 1897). Indeed, African Hebrew soldiers are constantly reminded by their commanders and fellow soldiers of their otherness—as non-Jews, as black people, as residents of "periphery" (often used as a euphemism for

---

2　While conducting fieldwork, I was surprised to find that many community members were under the false impression that their youth serve in the IDF voluntarily, and some researchers have repeated this claim (Jackson 2013). However, according to the 1986 Defense Service Law, both Israeli citizens and permanent residents of military age are obligated to enlist. Former Interior Minister Avraham Poraz confirmed to me in an interview that IDF service is compulsory for the African Hebrews because they are permanent residents. To be clear: The African Hebrews do not volunteer for the IDF, though a few soldiers have opted to extend their service beyond the required 2–3 years in order to become officers.

3　In February 2015, 19-year-old Cpl. Toveet Radcliffe was found dead in a guard post on her base. After a short investigation, the IDF ruled that Radcliffe, who was raised in the AHIJ community, committed suicide. Her family and friends rejected this explanation and accused the army of covering up her murder. The IDF reopened its investigation in 2017 and again determined that Radcliffe took her own life, either willingly or accidentally as a result of mishandling her rifle. About the case, Nasik Immanuel Ben Yehuda, a member of the AHIJ's ruling body known as the Holy Council, told the media: "There's this nagging feeling that things would be different if Toveet wasn't a little black girl from Dimona, and that somehow she's less important than someone else. And something's wrong with that picture." Some community members said they would not send their children to the military because of how this case was handled, potentially upending an arrangement between the community and the state of Israel dating back to 2004. However, I'm not aware of anyone following through on this threat. See journalist David Sheen's reporting (Sheen 2018) for details about the case.

4　For a thorough comparison of how these ethnic and religious minorities navigate IDF service, see my unpublished master's thesis (Esensten 2015), from which this article is adapted.

backwards) Israeli towns like Dimona. They are excluded from parts of military life and are forced to transgress cultural taboos to fulfill their duties. Some become rebellious, some lose motivation. Yet in contrast to the feeling of shame that Du Bois identifies in black Americans who experience "double consciousness", most African Hebrews manage to hold onto their distinctive identity and even emerge with a greater sense of pride in themselves and their ability to overcome the many physical and spiritual challenges that military service presents. However, this pride does not always translate into a personal commitment to continue to adhere to their strict cultural guidelines after they are discharged.

## 2. Methodology

Between 2010 and 2014, I interviewed 30 African Hebrew soldiers during or shortly after they completed their mandatory IDF service. This article focuses on 14 of those soldiers who enlisted between 2009 and 2010, when the rules and regulations of the IDF were consistent. I used snowball sampling, starting with a group of young people I knew from my longtime association with the AHIJ and then asking them to recruit their friends to participate in my research.[5] Efforts were made to interview a diverse group who served in a variety of units. The group included 10 men and 5 women, one officer, and two combat soldiers. Interviews were conducted in person in the Village of Peace, the community's urban kibbutz in Dimona, and lasted between two and four hours.

My historical discussion of the AHIJ relies on government documents obtained during in-person visits to the Israel State Archives in Jerusalem, as well as on national and international media coverage of the AHIJ in English and Hebrew and first-person accounts written by AHIJ members. It is also based on interviews that I have conducted with founding members of the AHIJ and with current and former Israeli government officials from 2007 to 2015. Unless otherwise noted, direct quotations come from personal interviews conducted by me.

## 3. Brief History of the AHIJ

In order to contextualize the African Hebrews' participation in the IDF, a brief history of the AHIJ and its complex relationship with the state of Israel must be provided. The AHIJ emerged from a small Hebrew Israelite congregation located on the Southside of Chicago, Illinois, in the mid-1960s. Most members of the congregation, known as Abeta Hebrew Culture Center, were African Americans in their early 20s who had been raised as Christians but left the church as adults in search of greater spiritual fulfillment. The elders of the congregation, drawing on spiritual traditions dating back to the late nineteenth century, taught that African Americans were the direct descendants of the Biblical Israelites and that the Bible contained their history and foretold their redemption (Dorman 2013). As part of a process of "reclaiming" their true identity, Abeta members took Hebrew names and gathered to study the Old Testament and celebrate Shabbat and Biblical holy days in accordance with the laws of Moses (Gavriel HaGadol 1993).

Inspired by Marcus Garvey's Back-to-Africa movement and fed up with racial inequality in the United States, members of Abeta were already discussing the viability of an African migration when, in early 1966, a 26-year-old foundry worker and Abeta member named Ben Ammi (born Ben Carter in 1939) announced that God had spoken to him while he was meditating at home and instructed him to lead an exodus of African Americans to "the Promised Land." In later accounts of this crucial moment, Ben Ammi said it was the archangel Gabriel who had appeared to him in a vision that lasted approximately 45 s (Singer 1979). Since God did not provide Ben Ammi with a road map for the

---

5    I conducted my initial fieldwork in the AHIJ's Village of Peace in Dimona from September 2007 through November 2008 with the support of a George Peabody Gardner fellowship from Harvard University. Afterward, from 2010–2014, I lived in Israel while attending graduate school and made frequent trips to Dimona to continue my research and to report on the AHIJ for the Israeli newspaper *Haaretz*.

journey, Abeta members turned to the Bible for clues. They understood from Jeremiah 31:21[6] that they had to retrace the route by which their ancestors left Israel, that is, through West Africa, and from Ezekiel 20:34–35[7] that they would need to spend a period of time in "the wilderness" before entering the land of Israel (Gavriel HaGadol 1993).

The West African country of Liberia, which was established in the nineteenth century as a home for freed American slaves, seemed to satisfy the Biblical criteria. The AHIJ vanguard established an agricultural settlement in Gbatala, approximately 100 km from the Liberian capital of Monrovia, where they set up camping tents and built rudimentary structures. Over the next two and a half years, they worked to cleanse themselves of spiritual impurities and make the transition from, in the words of Ben Ammi, "the Negro of the ghetto to the sons and daughters of God" (Ben Ammi 1982). Some scholars have speculated that the African Hebrews intended to settle in Liberia permanently but left either because they could not support themselves or because, in November 1969, the Liberian government threatened to deport them for being "undesirable aliens" (Landing 2002). According to Ben Ammi, the group's original plan was always to continue on to Israel, and in early 1968, he visited Israel with another group member, Charles Blackwell. Ben Ammi returned to Liberia, but Blackwell—today known as Nasik ("Prince") Keskiyahoo—stayed to live on a kibbutz and report back on his impressions of the people and culture.

The second phase of the AHIJ vanguard's immigration began in August 1969, when five members of the Liberian contingent landed at Israel's Lod airport and requested immigrant status under the Law of Return, which affords Jews from anywhere in the world automatic Israeli citizenship. Surprised by the sudden appearance of black Americans claiming to be Jews, border authorities gave them provisional immigrant visas and sent them to the Ministry of Absorption, which settled them in the remote desert town of Arad. Another group of 39 African Hebrews arrived from Liberia in December. These would-be immigrants received three-month tourist visas, as well as apartments in Dimona and Arad and permission to work—an unusual, halfway absorption that reflected the government's uncertainty over what to do with them (Singer 1979).

By 6 March 1970, when Ben Ammi arrived in Israel with another group of African Hebrews, they were being processed as tourists. Israel's Chief Rabbinate had investigated their religious backgrounds and determined that they did not meet the criteria of *halakha* (Jewish law) to be considered Jews, namely because they did not have Jewish mothers. The African Hebrews were encouraged to convert to facilitate their absorption, but they refused because they felt no need to conform to the religious standards of white, Jewish rabbis (Singer 1979). Outraged by the discrimination that he claimed his followers were facing, Ben Ammi began to publicly express more radical elements of the group's ideology. During a 1971 press conference outside Zion Gate in Jerusalem, he accused Eastern European Jews of usurping the land of Israel and threatened to call forth "plagues tenfold and a thousandfold" if the African Hebrews—"the true descendants of the Israelites"—were not given residency, housing, and jobs (Landau 1971). He also claimed that millions of black Americans were preparing to "repatriate" to Israel and that they would soon overthrow the Israeli government and establish the "Kingdom of God", as foretold in Daniel 2:44[8].

In response to Ben Ammi's increasingly inflammatory rhetoric, the government stopped renewing the visas of African Hebrews already in the country and began denying entry to African Americans

---

[6]　"Set thee up waymarks, make thee high heaps: set thine heart toward the highway, *even the way which thou wentest*: turn again, O virgin of Israel, turn again to these thy cities" (emphasis added). (N.B. The AHIJ use the King James Version of the Bible, so verses from that version will be quoted here.)

[7]　"And I will bring you out from the people, and will gather you out of the countries wherein ye are scattered, with a mighty hand, and with a stretched out arm, and with fury poured out./And *I will bring you into the wilderness of the people*, and there will I plead with you face to face" (emphasis added).

[8]　"And in the days of these kings shall the God of heaven set up a kingdom, which shall never be destroyed: and the kingdom shall not be left to other people, but it shall break in pieces and consume all these kingdoms, and it shall stand for ever." In his first book, *God, the Black Man, and Truth*, Ben Ammi described the "Kingdom of God" as "the vehicle which will bring deliverance, hope and redemption unto all men" (Ben Ammi 1982).

whom they suspected of planning to join the group. Undeterred, the African Hebrews used a number of tactics to avoid deportation so that they might remain in the country and establish the "Kingdom of God". One of the tactics was to stage protests, both in Israel and the U.S. Another was to petition the Israeli courts for more rights in the country, though these efforts often failed. A third tactic was to renounce their American citizenship so they would become stateless and therefore ineligible for deportation. Beginning in 1973, on instructions from Ben Ammi, community members began appearing at the U.S. embassy in Tel Aviv to give up their citizenship. Government documents reveal that Israeli officials were furious with the Americans for allowing the African Hebrews to become stateless in contravention of international law. According to these documents, the Americans viewed the problems created by the group as an internal Israeli affair.

Over the next two decades, from the late 1970s to the late 1990s, the government and the AHIJ engaged in a war of attrition: Israel deported scores of African Hebrews, seemingly at random, while group members continued to sneak into the country, often posing as Christian tourists and overstaying their tourist visas. In August 1978, the Knesset, Israel's parliament, took a concrete step toward resolving the problems presented by the African Hebrews' settlement in Israel by appointing a special committee to study the situation. The committee, led by Knesset member David Glass, concluded that the African Hebrews should have been denied entry or deported shortly after their arrival but that the government had adopted an "ostrich-like policy" for fear of appearing racist and provoking international outrage. The committee argued that the government would have been justified in expelling community members early on but that "more than ten years after the arrival of the first group, we have missed the boat." The committee further recommended that the AHIJ be granted permanent residency status and allowed to build a settlement in southern Israel on the condition that they not allow more members to join from overseas (Glass et al. 1980). The government ignored these recommendations.

The relationship between the state and the AHIJ reached a nadir in the mid-1980s. Dov Shilansky, a deputy minister in the Prime Minister's Office, pledged that the entire community would be deported in the spring of 1984. He went so far as to compare them to a terrorist organization, telling the *Jerusalem Post*: "They are worse than the [Palestine Liberation Organization]. They want to take over" (Moriel 1984). The government did not act on Shilansky's threat that year, continuing a pattern of heated rhetoric followed by inaction. However, on 16 April 1986, security forces arrested 46 members of the community in a predawn raid on an orange plantation in Rehovot where they had been working without permits. Then, on 22 April, hundreds of Border Police officers surrounded the Village of Peace in Dimona to prevent the African Hebrews from marching to Jerusalem to protest the mass arrest and deportation of their members. The march was called off, and the community refers to the event as the "Day of the Show of Strength."

The African Hebrews continued to languish in a state of legal limbo through the late 1980s. Finally, in July 1990, Interior Minister Aryeh Deri announced that African Hebrews who held American citizenship would be able to obtain work permits, prompting many of those who had renounced their citizenship to reapply for it. Why Deri, a member of the ultra-Orthodox and rightwing Shas party, would extend these rights to the African Hebrews when previous ministers failed to act remains unclear. By September, 860 African Hebrews had received B1 visas allowing them to work. Two years later, Deri upgraded their status to temporary residency. In 1993, an Israeli-run school was built for the community in Dimona with money from a U.S. government grant.

The decision by then-Interior Minister Avraham Poraz to grant AHIJ members permanent residency in 2003 was a major turning point for the community, as it meant that after more than three decades their quest to establish a permanent home in Israel had finally been achieved. But with this achievement came a new challenge: serving in Israel's military.

### 4. "Digging Deeper" into Society: AHIJ Reasons for Accepting IDF Service

The African Hebrews maintain that they have been ready and willing to serve in the military since they arrived in Israel (Fishkoff 1993). In the late 1980s, a group of young African Hebrew men presented themselves at IDF headquarters in Tel Aviv and asked to be drafted into the army as volunteers. They were rejected, presumably because they lacked legal status in the country. In addition, persistent but unfounded rumors that the African Hebrews were colluding with Israel's Arab enemies also would have made army officials hesitant to draft them. On this point, the Glass report stated:

> To date, no contact has been discovered with any hostile group, for the purpose of actively injuring the State of Israel in any way. However, the very existence of a closed, mystic cult with a strong motivation and an ideology based on the assumption that the 'Black Hebrews' are the true masters of the Land, obliges the authorities to maintain a security alert regarding the activities of the cult. (Glass et al. 1980)

When Israel granted the African Hebrews permanent residency status in 2003, community leaders readily agreed for their members to be drafted. Interior Minister Poraz recalled: "We wanted it, they wanted it, it's good for everybody. I don't think that it would have been smart from an Israeli point of view to let them stay without military service." In fact, community leaders agreed that both male and female members of military age would be conscripted; this was a highly unusual arrangement, since the female members of every other non-Jewish group in Israel are automatically exempt from serving in the army. Today, some community leaders admit that in lieu of military service they would prefer that their daughters perform national service (*sherut leumi*), which involves volunteering in schools, hospitals, and other public institutions. The majority of those who do national service are young Jewish women from the religious Zionist sector who receive exemptions from the army on religious grounds. It is not clear why community leaders did not push for this option for their daughters in 2003.[9]

The decision by AHIJ leaders to send the youth of the community to the IDF rather than protest on any number of grounds, such as their pacifist views, appears to have been guided by several factors. The first was a desire to improve their standing in Israeli society. Like the leaders of the Druze community, who agreed to a "pact of blood" with the Israeli government in May 1956, Ben Ammi and his deputies had reached the conclusion that entering the IDF was the best way to prove their loyalty to the state and ensure long-term residency in their putative homeland (Frisch 1993). This, after all, is what they most hoped to achieve when they left Liberia in 1969.

Another motivation was a desire to improve their social and economic status. Sar Avraham Ben Israel, a "minister" and the father of the first African Hebrew soldier to enlist in 2004, explained that the community wanted to demonstrate its commitment to the security of the state and that, in return, they expected to receive greater government support. He explained: "We knew that there would be benefits to go and dig deeper into the society and become a part of the state, and that would assist us because we've been struggling for so long. We knew that that would help us, not so much that we needed that, but it would make things a lot easier." Thus, the African Hebrews saw their participation in the IDF as part of a transactional relationship with the state.

There was also a practical reason to join the IDF: to protect themselves. Sar Avraham explained that despite their moral opposition to armed conflict, the African Hebrews could not ignore the very real threats to Israel's security and, ultimately, to their own safety. He said: "We don't believe in no fighting and no killing. On the other hand, we're living in a country that's at war. It's surrounded by enemies, talking about annihilating [it]. So at that point you don't have a choice but to defend yourself." Sar Ahmadiel Ben Yehuda, the community's "minister of information", emphasized how

---

9　There is no practical political effort to change the status quo with regard to the military service of female African Hebrew soldiers at this time. AHIJ leaders and activists are currently focused on securing full Israeli citizenship for all community members and securing the permission and funding for a new housing development in Dimona to accommodate their ever-growing numbers.

closely the African Hebrews identify with the Israeli state when it comes to security matters: "We went through this dynamic back in the Gulf War with Saddam Hussein, and that first Scud missile being sent into Israel did not discriminate relative to whoever it would land upon. So any attack on the state of Israel is regarded as an attack on the community."[10]

Yet another factor guiding the decision to join the IDF, according to community leaders, was a feeling of societal obligation to pay their "debt to society" by contributing to the national defense. Sar Avraham explained: "We can't sit back and allow our brothers and sisters in the country to send their daughters and sons out there and sacrifice them on behalf of their nation and we don't give anything." When Sar Avraham's son, Oriyahu Ben Israel, was inducted into the IDF in 2004, Ben Ammi was quoted in the Israeli newspaper *Haaretz* as saying: "From now on we will be an inseparable part of Israeli society. Until now, other people sent their kids to protect us. Now it's time for us to pay our debt" (Hasson 2004). There is no denying that the African Hebrews conceive of their community as bound by the national social contract that requires non-Haredi Jewish Israelis to protect the homeland.

The African Hebrews also see their participation in the IDF as consistent with their spiritual commitment to preserving Jewish sovereignty—as opposed to Arab/Muslim sovereignty—over the land of Israel. Sar Ahdeev Ben Yehuda, the community's liaison to the IDF, framed this commitment in Biblical terms: "We identify with the state and with the Biblical covenant that this land, from the Nile to the Euphrates, was given unto the seed of Abraham. We don't see any other national entity having the spiritual mandate over this land." Sar Ahmadiel clarified: "We do believe that the autonomy over this land should be in the hands of Israelites. Our ideals are in harmony with African Hebrews, Israelites, Judaism. There's more connection there than on the other side of the equation." (Of course, the African Hebrews have said they would prefer to rule over the land of Israel themselves one day.)

Beyond these motivations, it appears that some community leaders were amenable to their children being drafted because they themselves had served in the American military or in law enforcement as young men. The U.S. Army, which was desegregated in 1948, attracted large numbers of African American recruits in the 1950s and 1960s, when the leaders of the AHIJ were coming of age (Sutherland 2004). After dropping out of high school in Chicago, Ben Ammi enlisted in the army and served for three years on a missile base in Illinois (Gavriel HaGadol 1993). Nasik Keskiyahoo, the first African Hebrew to settle in Israel in 1968, also left high school to join the service; he was a paratrooper in the 82nd Airborne Division and spent two years stationed in Germany. Nasik Shaleak Ben Yehuda served in the U.S. Navy for two years in Great Lakes, Illinois (Lounds 1981). Nasik Elkannon served in the Illinois National Guard and then in the army, first as a military policeman and later as a driver for a reconnaissance unit. Three other senior leaders—Nasik Rahm, Nasik Aharon, and the late Ahk (formerly Nasik) Ahmeshadye—worked as policemen before relocating to Israel.[11]

## 5. "We Are Sons of Peace": AHIJ Concerns about IDF Service

Although AHIJ leaders had compelling reasons for agreeing to IDF service as part of their status upgrade, their entry into the military was neither straightforward nor uncontroversial. Sar Ahdeev

---

[10]   Left unstated by AHIJ leaders, who tend to avoid any public discussion of the Israeli–Palestinian conflict, was the fact that the community has been directly affected by terrorism. Aharon Ellis, the first African Hebrew born in Dimona and a talented singer, was murdered by a Palestinian gunman along with five other people at a bat mitzvah at which he was performing in Hadera in January 2002. At Ellis's funeral, Sar Ahdeev, then known simply as Ahdeev, told *The Jerusalem Post*: "Aharon symbolizes the first generation of our children in the country. We are prepared to sacrifice much more for the State of Israel to remain here in peace." (*The Jerusalem Post* 2002) Ellis's family sued the Palestinian Authority for enabling terrorism and eventually received a settlement of some kind. (Family members were unwilling to provide details.)

[11]   The influence of the leaders' military and police training is evident in nearly every facet of the AHIJ, from the rigid leadership hierarchy and the expectation of total obedience to authority figures—and swift punishments for acts of disobedience—to the strict dress code and the clearly defined protocol governing members' behavior. For many years, African Hebrew leaders ran a program called Y.O.T.C. (Youth Officers' Training Course), which was based on the U.S. army's Reserve Officers' Training Course model. The purpose of the course was to teach their sons discipline and self-defense techniques, as well as to train them to protect Ben Ammi when he traveled in Israel and overseas. This program was discontinued when the youth began enlisting in the IDF, suggesting that the army was also seen as adequately fulfilling a specific communal need.

called it "one of the greatest challenges" facing the community since their exodus from the United States. Some families quietly questioned the rationale behind the decision, given the contradictions between their ideology and the exigencies of military service. For the AHIJ, death is taboo. Ben Ammi preached during his lifetime that death, no matter the cause, is an aberration, and that humans can achieve physical immortality through proper diet, exercise, and prayer (Ben Ammi 1994). Military service, and all that it entails, ostensibly challenges the AHIJ's most cherished values. Meanwhile, members of other U.S.-based Hebrew Israelite camps loudly condemned the AHIJ for collaborating with the "Zionist" regime in Israel and directly contributing to the oppression of the Palestinians, whose plight is sometimes compared to that of African Americans under the system of racial discrimination known as Jim Crow and to blacks under apartheid in South Africa.[12]

Community leaders readily acknowledge the contradictions of their participation in the IDF, even if they find it hard to justify without subtly questioning the integrity of their ideological project. Sar Ahdeev explained:

> We don't think anybody should [have to] place their children in the military. We are sons of peace, and that's a part of our vision, that we have harmony in society, peace, and love and brotherhood. But given the current realities and circumstances, we try to make our soldiers understand that there's things that we have to go through that may not necessarily be part of our vision.

According to Sar Ahdeev, community leaders discussed the decision at length and held orientations for soldiers and their parents to clarify their purpose for joining the military.

One of the leaders' initial concerns was whether the young recruits would be able to continue to practice their culture during the course of their service. Leaders and senior military officials met in the months leading up to the first enlistments in 2004 to discuss matters such as diet and dress. Initially, the army agreed only to allow African Hebrew soldiers to wear cloth boots, instead of the standard-issue leather boots, because they are vegan and eschew animal products. In response to soldiers' complaints that they were not able to fulfill other communal obligations while serving (such as observing Shabbat and holy days), Sar Ahdeev began supplying new inductees with a letter providing details about the community's lifestyle and holy day guidelines. Soldiers reported that some commanders read the letter and honored its requests, while others completely disregarded it.

In late 2013, after a handful of soldiers petitioned the army for greater leniencies, citing those given to members of other minority groups, the IDF recognized the community as one of its "special populations." Today, African Hebrew soldiers are exempt from active duty on Shabbat (because they fast from sundown Friday to sundown Saturday) and on all of the holy days, including holy days unique to the community like New World Passover. As vegans, they are eligible to receive extra salary (approximately 500–600 NIS per month) so they can supplement their diet with food prepared at home. In addition, since the Torah forbids wearing mixed fabrics, they are permitted to wear cotton uniforms at all times. Some individual male soldiers have also managed to get exemptions from shaving their heads and beards—an exemption given primarily to religiously observant Jewish soldiers—because doing so violates AHIJ customs.

Another of the leaders' concerns was that the youth would be exposed to negative influences and forbidden practices, such as smoking and eating meat, by their fellow soldiers. Sar Ahmadiel Ben Yehuda, the community's "minister of information", acknowledged as much: "We know that some of them will dabble and experiment in some areas that were forbidden and unknown to them, in many cases, but I always say my job is to be here when they come back to their senses." In fact, some adults within the African Hebrew community see the experience of IDF service as a spiritual test for their

---

[12]  These criticisms have appeared in different online forums over the years. It should be noted that U.S.-based Hebrew Israelites have denounced the AHIJ since they began leaving the U.S., both because they feel the AHIJ's exodus was premature—many are waiting for a divine signal to begin the journey to Israel—and because they rejected Ben Ammi's claim to messianic status.

children. Sar Ahdeev explained: "We knew going in there that our children were going to have to be rooted. We told them, this is your test, now Yah's [God's] gonna test you, and you're gonna be put on the line to see if you're gonna maintain your allegiance to what you know and your identity, or not."

On the other hand, African Hebrews also see IDF service as an opportunity to spread their doctrine. Just as Israeli society inevitably has an influence on the African Hebrews, Sar Ahmadiel said, so too do they have an influence on Israeli society. He recalled a conversation with Ben Ammi in 2004 in which they discussed how the media coverage surrounding their entry into the IDF, and especially their special needs as vegans, might encourage the Israeli public to explore healthier lifestyle choices. Veganism has become more popular in Israel in recent years, with an estimated 5% of the population keeping a vegan diet, and the African Hebrews take some credit for this trend (Frazin 2016). However, the African Hebrews do not actively proselytize while in the IDF, and to date there have been only a handful of Jewish Israelis who have officially joined the community.

The question of whether or not African Hebrews should serve in combat units was contentious, so much so that it appears to have driven a wedge between Ben Ammi and Asiel Ben Israel, the community's former "international ambassador." African Hebrew youth are discouraged from joining combat units by their parents and teachers, though they are not explicitly forbidden from doing so. Rather, they are taught that their purpose is to be, in the words of Sar Ahdeev, "a spiritual element in the army that would help to reinforce the IDF." Even before they joined the IDF's ranks, the African Hebrews viewed their contribution to the army, and to the state at large, as primarily spiritual in nature. In fact, they believe they are partially responsible for Israel's dramatic victory in the Six Day War. They point to the fact that the war was fought and won as the group's vanguard were starting their exodus from the U.S., thus activating a spiritual force that bolstered Israel's military campaign.[13]

The parents of African Hebrew soldiers fear for the safety of their children but are proud that they are serving their country and representing their community in a positive way. They honor their soldiers by praying for them, paying tribute to them during holy day celebrations and community gatherings in Dimona, and allowing them to eat for free at the community's restaurants. They believe that since Ben Ammi himself encouraged service in the IDF, it must be in the best interest of the community. Many are convinced that their soldiers receive divine protection because of who they are and what they stand for. One parent noted: "As long as they keep the umbilical cord connected to the throne [upon which Ben Ammi sits], they seem to be protected."

## 6. African Hebrew Soldiers in Their Own Words

African Hebrew youth tend to approach their military service with one of two attitudes: there are those who see it as an opportunity to give back to their community and country while gaining new skills, and there are those who see it as a waste of time and who would not serve if they were not obligated to do so. The success of their service often hinges on their own attitude and ability to cope with the daily stresses of military life. Among the thousands of Israeli youth who enlist in the IDF each year, these attitudes are certainly not unique to the African Hebrew draftees. Yet what differentiates the African Hebrews is that they tend to face physical, emotional, and psychological challenges that soldiers from other cultural or socioeconomic backgrounds typically do not. And, according to the soldiers, their superiors too often prove unable or unwilling to help them overcome those challenges—either because they lack expertise or maturity (commanders are often only a year or two older than the soldiers in their charge), they want to avoid the appearance of favoritism, or for more objectionable reasons like prejudice.

---

13　This claim recalls those made by Haredim regarding the contributions of ultra-Orthodox yeshiva students who defer their army service in order to study Torah. One rabbi explained: "The minute we are studying we strengthen the connection for the man at the front. He has his gun, his tank or a plane, we strengthen him because we have a connection with . . . the Divine Creator . . . You can see it in the Six Day War or the Yom Kippur War." (Stadler and Ben-Ari 2003)

Below I present and analyze excerpts from my interviews with 14 African Hebrew soldiers who enlisted between 2009 and 2010. The branch of the military in which the soldiers served is indicated in parentheses, along with their main role during their service; all names have been changed to protect the soldiers' privacy.

## 6.1. Culture Shock

The first wave of African Hebrews to enlist in the IDF did not have older siblings or relatives who could tell them what to expect. Many of the soldiers interviewed *did* benefit from having family members who had already gone through the IDF, though some did not. In addition, as 11th grade students these soldiers participated in *Gadna*, an IDF program that simulates the military experience on actual bases. Yet despite this preparation, several soldiers said they experienced "culture shock" when they started basic training. For example, Yehoshua (Air Force, driver) recounted: "Before the army, I didn't have any Israeli friends. It was just us [African Hebrews]; we were raised inside of a bubble." In the army, he said, "it was like I was being opened to another world, another society." Sharing close quarters with unfamiliar Israelis made Anayah (Army, communications) uncomfortable. "When I first got in, I hated it", she said. "Being with a bunch of people I didn't know, crowded together in the same room. And then you had to take showers with all these people. That's something I wasn't used to." These complaints about military life reflect a normal response to the stresses of communal living with strangers.

While senior members of the AHIJ are convinced that the community is widely known and beloved all over Israel, African Hebrew soldiers' own testimonies suggest otherwise. Their experiences indicate that the AHIJ is actually little known outside of the Negev. Israelis may have heard about the community in passing or seen African Hebrew entertainers on television, but in general their knowledge is superficial. Thus, African Hebrew soldiers are subjected to endless questions about who they are, where they come from, and what they believe. Tifoosah (Air Force, airplane technician) explained: "Israelis want to know everything about your life ... and they're going to tell you everything about them, so I couldn't stay [closed in] too long." She said her commander asked her to explain to her comrades about her background and community. She told them that her parents were Zionists but did not elaborate on their beliefs.

Sometimes these interrogations felt intrusive to the African Hebrew recruits. Amiel (Army, communications) said that his fellow soldiers knew "only the bad things" about the community, referring to cases of physical and sexual abuse that had been reported in the Israeli press, which he called "propaganda."[14] He explained: "It was more so agitation than actual genuine interest in the community itself, or in who I am, and in what my background is." In contrast, Avikhiel (Army, artillery) said that he did not clash with any of the soldiers he trained with, attributing their cohesion to the format and intensity of the training:

> I think the essence of combat basic training is to make sure you get along. There were all kinds of exercises that were designed to make us unite. I would say after basic training I had some pretty close friends, and even today we still keep in contact, even though we went to different divisions and units.

## 6.2. Racism

A number of soldiers reported that they were the only African Hebrews—and sometimes the only black people—on their bases and that they encountered racism in various forms. Several soldiers said they could not tolerate the racially insensitive language that some soldiers and commanders freely used in their presence. For example, Yermiyahu (Army, civil administration) said his unit commander

---

[14]　See Esti Aharonovitch's reporting in *Haaretz* (Aharonovitch 2009). An English translation of that article is available here: http://www.andrewesensten.net/translations/terrible-secrets-african-hebrew-israelite-community/.

called him *cushi*, a derogatory Hebrew word for black people, at morning roll call because he was not dressed properly. He said he was so incensed that he slapped the commander across the face and was sentenced to three weeks in military jail. (According to Yermiyahu, the commander was also disciplined.) Another African Hebrew soldier said he slapped a soldier after warning him several times not to call him *cushi*.

Nekmadiyah (Army, combat engineer) felt his commander singled him out to sweep and clean bathrooms on his base because of his skin color. He recalled: "I went up to him and was like, 'There's something about me you don't like. Let's sit down and talk about it.' He's like, 'No, I've got nothing to say to you, *cushi*.' I was like, whoa, this is my commander, and he's taught to relate to all his soldiers and try to avoid confrontation with them." Nekmadiyah said he lodged a formal complaint over the incident, and his commander was moved to another platoon.

Amiel said the racism he experienced in the army was "heavy", recounting one situation in which his commander told him to "go back to your country", not knowing he was born and raised in Israel. He said he sometimes talked back to his commanders and disobeyed orders because "I refused to get to the point where I could let somebody feel that because I'm a black person, or because I'm from the [AHIJ] community, then they're allowed to treat me however they want to treat me, tell me whatever they want to tell me, and I'm supposed to march to the beat of their drum. That's not how I operate." Yet from his perspective, Ethiopian soldiers suffered worse treatment by their fellow soldiers and commanders: "I saw it much worse with the Ethiopians. Ethiopians won't defend themselves. The majority of them just take whatever is given to them and they accept that kind of attitude for some reason. I wasn't going for it."

In contrast to these testimonies, Elroi (Army, human resources) said that he had not been subjected to any racist comments or slights in the IDF and that "people are actually a lot better than what I expected." Likewise, Avikhiel said he did not encounter overt racism. In fact, he said he was afforded respect because he carried himself differently from other soldiers: "The moment you show them that you're different, that you're special, they show more respect. When you start breaking down the standards and principles that the Kingdom stands on and for, they show respect." Thus, race was certainly a factor in the African Hebrews' experience in the IDF, but the extent to which it mattered varied from solider to soldier.

*6.3. Dietary Challenges*

The African Hebrews adhere to a strict vegan diet based on a selective reading of the Torah, and many soldiers complained about the lack of vegan food on their bases and in the field.[15] Yanah (Army, electronics) recounted: "Basically there was only salad for you to eat, and when my parents sent me food they didn't want to put it in the refrigerator. They said it's 'special privileges,' so I had to go off on them about that. I have to eat just like everybody else." She said her commander told her that she was not familiar with the AHIJ and asked her why she was being so difficult. "We went back and forth all the time", Yanah said.

Several soldiers said they could not even enter the dining halls on their bases because they did not feel comfortable there, specifically because they could not stand the smell of meat. During basic training, Matanah (Army, communications) told her commander that she couldn't eat in the cafeteria because the smell made her sick. The commander responded by having a box of cucumbers and tomatoes sent to her room. After that, Matanah began bringing her own food and storing it in the kitchen on her base. However, she said a rabbi on the base put a stop to that because he did not consider

---

[15]  The AHIJ diet is based on a literal reading of Genesis 1:29: "And God said, Behold, I have given you every herb bearing seed, which is upon the face of all the earth, and every tree, in the which is the fruit of a tree yielding seed; to you it shall be for meat." They consider this to be the ideal human diet because God gave it to Adam and Eve before expelling them from the Garden of Eden.

the food she brought from home to be kosher. Other soldiers reported skipping meals; one mother said that her son developed an ulcer because he was not eating properly.

Lack of access to vegan food was an acute hardship for combat soldiers who remained in the field for weeks at a time during basic training. Since they could not eat the canned tuna that was provided to them, they subsisted primarily on corn, pickles, and bread. One African Hebrew soldier who enlisted in 2005 as part of a Nahal unit said he turned in his gear and threatened to quit unless he received more nutritious food. Years later, African Hebrew combat soldiers still struggle with this aspect of their training. Avikhiel said he adapted out of necessity but that it was incredibly stressful.

### 6.4. Inflexible Standards and Commanders

Before the IDF recognized the AHIJ as a "special population" in 2013, African Hebrew soldiers had to individually petition their commanders for exemptions and privileges that they felt they should have been granted automatically for cultural reasons. Of course, some commanders were more understanding than others, and this lack of consistency was a major source of frustration. As one soldier put it: "In the army it's hard to readjust to that system, especially if you don't have a commander that cooperates with you, even on the smallest things." As a result, some of the African Hebrew soldiers were disciplined for violating rules while others compromised on their principles in order to avoid punishment.

Yehoshua refused to shave his head, like all non-Haredi recruits are expected to do, because African Hebrew men are not supposed to shave their heads or beards in observance of Leviticus 19:27.[16] He said his superiors threatened to send him to jail, but "once they saw that I was one of those strong spirits that couldn't be broken, they gave in." Likewise, Elroi was brought to military trial for refusing to crop his hair. In this case, the judge excused him on religious grounds. However, later in his service, he was told that he would not be able to participate in an officer's training course unless he cut his hair. He agonized over the decision and, in the end, decided to conform to army guidelines so that he could advance in rank. He recalled: "My spirit wasn't right for that first week. I was messed up. I think that was the hardest part of the course."

Other soldiers said they clashed with their commanders over the strict military dress code. Yermiyahu was moved to another base after refusing to sign a form forfeiting his status as a vegan soldier—and the extra salary that comes with it—because he chose to wear leather boots. He explained: "We're vegans because of health reasons, so the leather boots are not that big of a deal. After arguing with the commander for forever, I told him I'm not going to sign so he ended up moving me, basically over boots." Matanah said that she often ran away from military policemen who noticed that she had modified her uniform by sewing "cords of blue" on the corners of her pants, in accordance with the African Hebrews' observance of Numbers 15:38 but in violation of the IDF dress code.[17]

### 6.5. Sabbath Observance

Another challenge that military service presented to the African Hebrew soldiers was how to observe Shabbat on their bases. During basic training and sometimes afterward, depending on their jobs, soldiers are required to "close" Shabbat on base. In general, African Hebrew soldiers objected to having to remain on their bases over Shabbat because they fast from sundown on Friday until sundown on Saturday every week. Some African Hebrews convinced their commanders to allow

---

[16]　"Ye shall not round the corners of your heads, neither shalt thou mar the corners of thy beard."

[17]　"Speak unto the children of Israel, and bid them that they make them fringes in the borders of their garments throughout their generations, and that they put upon the fringe of the borders a ribband of blue." It should be noted that conflict over the dress code is exacerbated by the fact that African Hebrew soldiers have different personal taste and levels of conviction. For example, some wear the standard-issue polyester service uniform (*madei aleph*), while others request permission to wear the cotton field uniform (*madei bet*) and cloth shoes. In addition, some male soldiers wear *kippot* while others do not. One community leader faulted the soldiers themselves for failing to consistently uphold their cultural standards while in the military, thus allowing commanders to ignore some of their requests on the grounds that they are optional practices.

them to remain in their rooms as a compromise. A few soldiers said they were forced to eat to sustain themselves. Elroi explained: "I was fasting, but I allowed myself to eat a little something here and there . . . to maintain my physical strength."

Arguing that they would not be able to perform guard duty, drive, or do any other task that would require physical exertion or the use of electricity, some soldiers petitioned their commanders to let them leave for the weekend. Nekmadiyah said his commander refused to let him leave one Shabbat, so he left anyway. As a consequence, he was given a military trial and received a sentence of 28 d on base. He served the sentence but said he later appealed it and received "credit" for having been on active duty for those 28 d. Tifoosah said she fainted one Shabbat and was taken to the hospital. Subsequently, she was allowed to go home every weekend.

## 6.6. Economic Hardships

The IDF offers financial aid to needy soldiers, but the application process involves numerous home visits and large piles of paperwork. Yedidiel (Air Force, logistics) said the process dragged on for months and that he did not receive the aid he needed. "My family was struggling; my mother was sick", he said, noting that he was the head of his household during his service and had to juggle the military and his familial responsibilities. "I just felt like, as a soldier, I should have gotten more help", he said. In an unusual display of compassion, Yedidiel said his superiors gave him money out of their own pockets from time to time and allowed him to leave his base early so he could work at an auto supply store in Be'er Sheva.

Yanah appealed to her commander for housing, explaining that she did not have her own bed at her parent's crowded house in the Village of Peace. The commander did not follow through on her request. Yanah recounted: "They're supposed to check if you need something in your home, if your parents need help. At the time, I was sleeping on the couch and I told them this. They were supposed to help me get housing, but they didn't help with anything." She added: "I could have been a good soldier if they had treated me right."

## 6.7. Outsider Status

A few soldiers described problems that arose due to their outsider status as non-Jews without Israeli citizenship. For over a year, Tamara (Army, communications) served in a unit that handled sensitive communications for the Israeli president's residence in Jerusalem. After a year in this unit, she was suddenly removed. "They told me something was wrong with my [security clearance] and that I had to leave the room", she recounted. "It was terrible because I was supposed to become a commander and it happened right during the process. It was embarrassing." Her commander told her that she was removed from the unit because she did not hold Israeli citizenship. After that, she said, "I didn't have a job in the army so I was just coming to the base every day doing nothing."

Matanah said she dreamed of being a combat soldier but decided against it after learning that Jewish soldiers and non-Jewish soldiers who fall in the line of duty are buried in separate sections of the cemetery, per Orthodox Jewish custom. She said: "It got to me big time. Just to know the fact that I'm good enough to fight, but at the end of the day this country don't give a damn about me." This objection may be rooted in Matanah's realization that the IDF is a thoroughly Jewish institution that enforces Jewish religious law in many areas, including burial, and that excludes her because she is not considered Jewish.

Several African Hebrews participated in a course designed to "enhance the immigrant soldiers' ties to the State of Israel and the Jewish people and to help them integrate into society" (IDF Spokesperson's Unit 2010). The course, called *Nativ* ("path"), covers topics in Israeli history and Jewish religious practice, and soldiers are given the opportunity to convert to Judaism at the end. Few if any African Hebrews have actually gone through with conversion. Several soldiers explained that they took the course to get a break from their regular army duties. However, some African Hebrews said they did not feel welcome in the class. Anayah said the instructors dismissed her and the other African

Hebrews in the course because they refused to stay on the base over Shabbat as required. However, the soldiers believe they were dismissed because they repeatedly challenged the instructors on matters of Biblical interpretation.

*6.8. Combat Service and the Israeli–Palestinian Conflict*

Although African Hebrew soldiers said they are dedicated to defending Israel and securing its borders, they admitted to having conflicting emotions about their involvement in an intractable struggle that has resulted in thousands of deaths on both sides of the Israeli–Palestinian conflict. Amiel described the tension this way: "I'm in full favor of Israel, but that does not mean that I accept the fact that what we're doing to the Palestinians is right. It's been enough bloodshed, it's been enough wars, and we have enough mothers who are grieving over their children, both from our side and from the Palestinian side." Several soldiers who qualified to serve in combat units said they found ways to avoid being placed in those units because they do not believe in fighting Arabs and because they were afraid of being put on the front lines in a war.

On a more personal level, some African Hebrew soldiers struggled to justify their loyalty to a state that has treated their parents and grandparents—and other dark-skinned people—less than hospitably in the past. A handful of soldiers spoke about situations in which they found themselves defending or quietly sympathizing with the non-Jewish recipients of Jewish soldiers' abuse. While stationed along Israel's border with Egypt, Nekmadiyah came to the aid of a Sudanese refugee who had crossed the border illegally and was receiving poor treatment at the hands of the Border Police. He jumped out of his vehicle when he saw the man being kicked and demanded that the beating stop. He said he spoke to the man and clipped the straps that were cutting into his wrists. When the officer at the scene reprimanded Nekmadiyah for becoming involved, he responded that he was only doing what was right and threatened to contact the United Nations if the officer attempted to formally reprimand him.

In general, African Hebrew soldiers tend to see themselves as disconnected from the Israeli–Palestinian conflict since they are neither Arab nor fully Israeli. Yet they said they feel solidarity with their fellow soldiers. As an example, Shekaniyah (Air Force, driver) was so moved by the plight of Gilad Shalit—the Israeli soldier who was abducted by the Palestinian militant group Hamas militants in a cross-border raid in June 2006—that he wrote a song about him. In "Lost", Shekaniyah sings from Shalit's perspective: "I'm so far from home/ I'm trying to get back to where I belong." He performed the song at community functions, as well as for Shalit's parents, Noam and Aviva Shalit, in Jerusalem before Shalit was released in October 2011 as part of a controversial prisoner exchange with Hamas.

*6.9. Double Consciousness*

The question "Do you feel Israeli?" elicited different responses from the soldiers. Some differentiated between a cultural identity (African Hebrew) and a national identity (Israeli). A few acknowledged that they had indeed picked up certain traits associated with the Israeli character but that this did not make them Israelis, per se. This psychological tension, whether explicitly acknowledged by the soldiers themselves or simply hinted at by their comments, is a major feature of the experience of the African Hebrews in the IDF. Much like Du Bois's concept of "double consciousness,", the soldiers perceived themselves as having two competing identities and to be straddling two different, seemingly incompatible worlds.

But are African Hebrew-ness and Israeli-ness truly incompatible, or have the soldiers simply internalized the us-versus-them mentality of their parents? For Yehoshua, the important distinction is not between cultural and national identity but between cultural and *religious* identity. He said he sees himself as Israeli because he was born and raised in Israel. Moreover, he said he feels no need to convert to Judaism because "I believe in the *Tanakh* [Hebrew Bible] just like [Jews] believe in it. But I have my own Kingdom *minhagim* [traditions] that I believe in as well. Ain't no need for me to adopt nobody else's identity because I have my own."

To the question of whether IDF service made him more Israeli, Shekaniyah responded in a way that suggested he found the two incompatible: "No, not at all. I'm not easily influenced because I adore my lifestyle. I like everything about it." Likewise, Nekmadiyah said he decided before basic training that he would not be influenced by the Israelis he encountered in the army and that he was mostly successful in this endeavor: "Before I went in the military here, I was like, 'Man, I will never be like them, I will never blend in.' It just made me get to know them. But I didn't forget where I came from or what my culture really was." Maskeelah echoed this idea, saying:

> You become different the moment you put this green uniform on. You're a symbol for something bigger than yourself. And wherever you are, you are representing Israel, the whole armed forces. I'm a symbol for Israel, but I'm a symbol for my nation, that's why I wear my cords [of blue]. I told them, you're not gonna turn me into you. You're not gonna make me be the person that you are.

Even as the soldiers denied becoming more Israeli, they asserted that they deserved Israeli citizenship for serving in the IDF. Legally, most African Hebrews were American citizens at the time the interviews were conducted. (Soldiers who do not hold Israeli citizenship can apply for it after completing half of their mandatory service.) Yermiyahu argued that the African Hebrews are more worthy of citizenship than others who do not contribute to the defense of the state:

> We trace our roots back to Judah and we consider ourselves Jews. According to law, we should get citizenship. And just the fact that we've been here for 40 years ... who doesn't get citizenship after living in a country for 40 years, serving in the army, contributing to everything? We contribute more than the Haredim. The Haredim are citizens; they don't even do the army. We do the army. We've been here for forever. Yeah, I think we deserve citizenship.

An aspiring politician, Amiel said he applied for his citizenship and has encouraged other soldiers to do so: "It's something that we deserve. It's not like [somebody's] doing us a favor by giving us citizenship. This is something that we earned, something that we deserve, so let's go get it."

### 6.10. Consequences of IDF Service

Several soldiers acknowledged that they gained important skills while serving in the IDF. For example, Shekaniyah said he benefited from having to speak Hebrew every day: "My Hebrew definitely got better, because I was a person who didn't like to speak Hebrew. If you didn't know English, I didn't want to talk to you. Now it's better and I can hold conversations and really express myself in Hebrew." Other soldiers said they learned how to be more assertive. Tamara explained: "It made me more aggressive in life, in getting what I have to get. Coming from the community, we are usually more polite, more laid back. We usually don't get aggressive, but I realized that that's how Israelis work."

Another benefit of serving in the IDF was that it allowed African Hebrews to expand their small social circles. Yehoshua said: "The good thing about [the IDF] is it takes a lot of people from different areas and cultures and brings them together and it makes them become one. One of my friends was a Russian, one was an Iraqi, another one was a Moroccan. We all [came] from different worlds, different cultures, but we all were so close." Yehoshua and his friends remained in contact after their service and celebrated Passover together one year at the Sea of Galilee.

As a result of the close bonds they formed in the IDF, African Hebrew soldiers said they also influenced their comrades' behavior and eating habits. Avikhiel reported that his comrades volunteered to fast with him one Shabbat, and a number of soldiers took pride in having convinced other soldiers to try to eat only vegan food for a day or two. It is not uncommon for African Hebrew soldiers to invite their comrades to visit the Village of Peace and celebrate holy days with their families. Thus, as their leaders hoped, African Hebrew soldiers have become important ambassadors for the community in Israel.

On the other hand, Matanah said that some of her peers are easily influenced by others and "forget who they are" once they enter the IDF. She said some of her friends began self-identifying as American, eating meat, smoking, and engaging in other behavior that is prohibited in the community. In one case, an African Hebrew woman met her future husband in the *Nativ* course and, as a result, decided to give up her cultural traditions. Another soldier decided to leave the community after completing his service because he is gay and homosexuality is forbidden. He said he felt empowered to make decisions on his own, without having to consult community leaders, after participating in an officer's training course. Undeniably, service in the IDF leads a certain percentage of African Hebrews to renounce their teachings and leave the community.

## 7. Conclusions

The Israel Defense Forces was conceived as a tool to unify Jewish immigrants from widely different origins and cultures who streamed into Israel shortly after the founding of the state in 1948. In many ways, the IDF was supposed to be the place that newcomers would learn to be Israeli. However, just like every other institution in Israel, the IDF was and still is fractured along ethnic and religious lines. The experiences of third-generation African Hebrews in the IDF presented here suggest that the Israeli army is an imperfect tool of acculturation for immigrants and minorities, especially those who are not Jewish.

The IDF did not succeed in creating Israelis out of African Hebrew soldiers interviewed for this project because they were enculturated from a young age with a unique set of values and beliefs about themselves and others that no state institution could unteach. And yet, despite this stable internal identity, the experience of serving in the IDF led some African Hebrew soldiers to decrease or drop their practice of the unique AHIJ lifestyle. There are no hard numbers of this phenomenon, and those who have left the community are generally hesitant to speak about their experiences. Yet community leaders are clearly alarmed by the number of veterans who leave. Sar Ahdeev said that approximately 40 percent of the youth who have gone into the army have had difficulty adjusting to post-army life, with some dropping out of the AHIJ structure altogether. He explained:

> Some have not adjusted; some have adjusted well. The majority, 60 percent or so, have adjusted. They haven't shown any signs that they're not adjusted. The other ones, we've had some that have fallen by the wayside in regards to maintaining their identity that they were born into and their cultural traditions. They've abandoned those things.

In response to adjustment difficulties and dropouts among African Hebrew veterans, Ben Ammi initiated a program called Yah's Exemplary Soldiers (YES) to help soldiers "readjust" to life after the military. According to Sar Ahdeev, the goal of the program is to better prepare the youth for the challenges of army life through the mentorship of current soldiers, as well as to provide them with emotional and psychological support during and after their service. Future studies on the AHIJ should consider the connection between army service and defection from the community, as well as its impact on university enrollment and professional advancement. (An informal survey of soldiers interviewed for this project revealed that fewer than half of each class had pursued post-secondary studies at least two years after completing their service.)

As an ethnic and religious minority group that has had an often contentious relationship with the state, the African Hebrews find themselves in a socially and politically compromised position similar to that of Israel's Arab citizens. Like Israeli Arabs, the African Hebrews have clashed with their Jewish neighbors and been accused of being a fifth column. They may never be fully understood or accepted, and this is perhaps the great irony of their emigrationist project: they left the U.S. because they resented their second-class status as black citizens of a country with a white supremacist power structure. Yet in their adopted home of Israel, they are *still* second-class citizens, both because they are black and because they are non-Jews in a country that defines itself as "the nation-state of the

Jewish people."[18] The death of Ben Ammi in 2014 sent shockwaves through the community but did not have an appreciable impact on its relationship with the Israeli government. It remains unclear if the community's accommodationist approach to the state will continue, or if the African Hebrews will, like the Haredim, withdraw further into their insular community out of fear of greater assimilation.

**Funding:** This research received no external funding.

**Acknowledgments:** I want to thank the leaders of the AHIJ, especially Ben Ammi Ben Israel z"l, for allowing me to freely conduct research for this project, and to all of the soldiers who shared their experiences in the IDF with me. I also want to thank Uriya Shavit of Tel Aviv University for encouraging me to explore this topic and serving as the advisor for my MA thesis, from which this article is adapted. I am grateful to Jonathan Esensten for reviewing drafts of this article and making many helpful edits and suggestions. I dedicate this work to the memory of Aturah Nekamah Baht Israel (1948–2015).

**Conflicts of Interest:** The author declares no conflict of interest.

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
