# Peer review of "Yah’s Exemplary Soldiers: African Hebrew Israelites in the Israel Defense Forces"

_religions, doi:10.3390/rel10110614_

Round 1

Reviewer 1 Report

A very interesting subject about a little known sect/minority. Though not cognizant to Du Bois' work (the reader seems to be expected to know it), the notion of 'double consciousness' does not strike me as particularly original or contributing much to the argument made in this study. Identities are always a complex mix of components even for natives, some hardly reconcilable ones in theory, but most people manage to steer an uncomfortable course though the maze. In this sense, the AHIJ's case is hardly special.

As the final lines of the conclusion indicate, it will be very interesting to see which way the community evolves, towards greater integration in secular Israeli citizenry or towards isolation.

Contrary to the claim at line 658 that there is a scholarly consensus (no reference, but a consensus is never to be taken as face value) 40% of AHIJ veterans decrease or drop their practice (line 664) it could be argued that the IDF is doing a pretty good job at acculturating this particular minority. Of course, finer study of the difference between decreasing practice and total drop out would be most helpful, but indeed complicated. Comparative data with the Druze and other minorities would provide a better grounded conclusion.

 Integrating elements from Du Bois' work may help too.

Reviewer 2 Report

This is a brilliant contribution which should be required reading for young IDF commanders.

I am someone who is interested in (and thought myself knowledgeable) Israeli minority politics, but this essay has taught me much that I did not know.

There are only a few minor issues and comments for reflection.

Lines 54-55, do you citations for similar work about Arab Israeli IDF service?  I would be curious to know...

Line 72, please define "snowball approach" for those readers outside your disciplinary niche.

Line 204 – is there any practical political effort to change this in 2019?  I am struck by the many ways that the African Hebrew community straddles the chasm between traditionalism / orthodoxy  and non-Jewish status.  (and later in the essay, since when are vegetables not kosher?  Do strict rabbis have to watch the cucumbers grow from seeds to label them as such?)

231 – Performing in what capacity?  Some explication here would be welcome.

423 – "Go back to your country" is a notorious racist cliche known to American blacks, and which has been very recently dredged up on the Twitter-feed of a certain high-ranking American politician (who has also made claims represent Israeli interests and has labeled critics as anti-semitic)

589-595 This quotation is a pithy statement about the several different Judaisms of Israel in a nutshell.  It could be equally applied to Samaritans or others.  I wish the concept of Israeli religious pluralism would be embraced by the mainstream in Israeli politics.

637 there is an extra space in this line.
